# Characterization of Non-Invasively Induced Post-Traumatic Osteoarthritis in Mice

**DOI:** 10.3390/antiox11091783

**Published:** 2022-09-09

**Authors:** Fazal-Ur-Rehman Bhatti, Yong-Hoon Jeong, Do-Gyoon Kim, Ae-Kyung Yi, David D. Brand, Karen A. Hasty, Hongsik Cho

**Affiliations:** 1Department of Orthopedic Surgery and Biomedical Engineering, University of Tennessee Health Science Center, Memphis, TN 38163, USA; 2Osong Medical Innovation Foundation, Cheongju 28160, Korea; 3Department of Orthodontics, The Ohio State University, Columbus, OH 43210, USA; 4Department of Microbiology, Immunology and Biochemistry, University of Tennessee Health Science Center, Memphis, TN 38163, USA; 5Department of Medicine, University of Tennessee Health Science Center, Memphis, TN 38163, USA; 6Research Service, Memphis VA Medical Center, Memphis, TN 38104, USA; 7Department of Orthopedic Surgery, Campbell Clinic, Memphis, TN 38104, USA

**Keywords:** post-traumatic arthritis (PTOA), mouse model, behavior test, reactive oxygen species (ROS), matrix metalloproteinases (MMPs), type II collagen (CII), non-invasive mechanical loading (ML), small animal imaging

## Abstract

The pathophysiology of post-traumatic arthritis (PTOA) is not fully understood. This study used non-invasive repetitive mechanical loading (ML) mouse models to study biochemical, biomechanical, and pain-related behavioral changes induced in mice. Mouse models reflected the effects of the early stages of PTOA in humans. For the PTOA model, cyclic comprehensive loading (9N) was applied to each mouse’s left knee joint. ML-induced biochemical and molecular changes were analyzed after loading completion. Cartilage samples were examined using gene expression analysis. Tissue sections were used in subsequent OA severity scoring. Biomechanical features and pain-related behavior were studied after 24 h and three weeks post-ML sessions to examine the development of PTOA. The loaded left knee joint showed a greater ROS/RNS signal than the right knee, which was not loaded. There was a significant increase in cartilage damage and MMP activity in the mechanically loaded joints relative to non-loaded control knee joints. Similarly, we found a difference in the viscoelastic tangent, which highlights significant changes in mechanical properties. Biochemical analyses revealed significant increases in total NO, caspase-3 activity, H_2_O_2_, and PGE2 levels. Gene expression analysis highlighted increased catabolism (MMP-13, IL-1β, TNF-α) with a concomitant decrease in anabolism (ACAN, COL2A1). Histopathology scores clearly indicated increases in OA progression and synovitis. The gait pattern was significantly altered, suggesting signs of joint damage. This study showed that biomechanical, biochemical, and behavioral characteristics of the murine PTOA groups are significantly different from the control group. These results confirm that the current mouse model can be considered for translational PTOA studies.

## 1. Introduction

Joint tissue injury leads to progressive damage to the articular joint, leading to post-traumatic arthritis (PTA) [1]. In total, 20–50% of individuals subjected to joint trauma develop post-traumatic osteoarthritis (PTOA) that results in damage to cartilage tissue and other tissues of the articular joint. Moreover, PTOA accounts for 12% of all osteoarthritis (OA) cases [2,3]. An acute knee injury such as trauma or excessive cumulative stress initiates biomechanical changes as well as biochemical cascades such as inflammation and metabolic imbalances of tissue turnover that can lead to secondary OA [4,5]. This PTOA is characterized by damage to multiple components, such as articular and meniscal cartilages, synovium, and subchondral bone. Evidence from both pre-clinical and clinical studies reveals that early intervention can moderate these biochemical changes and that altered joint biomechanics may slow the progression of joint degeneration [6].

Animal models dramatically shorten the time required to develop PTOA. A 3- to 6-month-old mouse is equivalent to a 20- to 30-year-old human [7]. To study the mechanisms of PTOA initiation and progression relevant to human disease, animal models must reproduce specific traumatic injury conditions seen in patients. The PTOA model can be categorized into invasive or non-invasive models. Invasive models involve surgical or chemical induction to develop OA. They require a great deal of intervention and mostly represent the acute phase of traumatic injury. Non-invasive mouse models have been used to mimic PTOA in humans to follow the progression of disease after traumatic injury [8,9]. These models involve subjecting a joint to a repetitive mechanical load. This model’s advantages compared to invasive models include being non-invasive, no skin disruption, and mimicking early adaptive response to injury in humans. Mechanical loading (ML) of the mouse knee serves as a model of PTOA and can faithfully mimic injury conditions like those in the early stages of human OA [9]. The pathology of PTOA involves three phases: the immediate phase following mechanical injury; the acute phase displaying apoptosis and inflammation; and the chronic phase characterized by dysfunction and pain of the joint [10]. Biochemical changes occurring after trauma mainly depend on whether the impact is of low or high amplitude [11]. The most common irreversible biochemical change is the breakdown of the cartilage matrix due to a disruption of proteoglycans that expose the underlying collagens maintaining the stability of the cartilage matrix. Type II Collagen (CII) is the main collagen affected in the hyaline cartilage [12]. This causes necrosis of chondrocytes residing within the cartilage matrix [13]. Various molecular factors responsible for cartilage tissue catabolism include the production of matrix metalloproteinases (MMPs), reactive oxygen species (ROS), and inflammatory cytokines [14,15,16]. ROS are released from storage in the mitochondria after excessive mechanical stress in the knee joint. The release of these destructive metabolites can result in chondrocyte death and matrix degradation [5]. Previously, we developed an in vivo detection technique for visualizing early damage to cartilage using a fluorescence-labeled monoclonal antibody (Mab) that binds to CII [17]. This method has the advantage of detecting damaged cartilage in the mouse knee joint.

A functional characterization, such as pain-related behavior, and previous conventional methods of studying PTOA are essential for the translation of preclinical studies to clinical OA therapy. Therefore, we utilized this PTOA mouse model to characterize many facets of arthritis from the macroscopic to those at the molecular level. Specifically, we provide previously unaddressed behavioral data related to the OA pain mechanism.

## 2. Materials and Methods

### 2.1. Animals

Mice (CBA-1, 12 weeks old, male, *n* = 16/group) were randomly divided into three groups (Figure 1): Control (Normal control without ML), Trauma (the immediate phase following the last ML session, which involved the cell apoptosis and joint inflammation), and PTOA (early OA: 3 weeks after the last ML) group (Figure 1). To reduce experimental variability (i.e., age, sex), and eliminate the need for extensive testing of the loading conditions necessary to produce PTOA of manageable severity, we used male mice of a particular age. Mice were provided with a healthy, pathogen-free environment ad libitum. All animal protocols and experimental procedures were approved by the Institutional Animal Care and Use Committee (IACUC) at the University of Tennessee Health Science Center (ethical protocol code: 17-055.0).

### 2.2. Non-Invasive Post-Traumatic Osteoarthritis (PTOA) Mouse Model

The PTOA was developed as described previously with slight modifications [18,19,20]. The left leg of each mouse was positioned within the ElectroForce^®^ 3200 (Bose Corp., Framingham, MN, USA) using a custom-made ML jig. The knee joint was positioned with the proximal tibia resting in the upper cup and the dorsiflexed ankle inserted into the bottom cup. Each loaded knee joint received 40 cycles of compressive loading at 9 N, totaling six successive sessions (3 times per week over 2 weeks).

### 2.3. Dynamic Mechanical Analysis (DMA) of the Knee Joints

The tibia was glued to the jig of ElectroForce^®^ 3200 instrument (Appendix A). The specimen was secured by preloading at 1 N on the loading jig. Next, compressive cyclic displacement (0.01 mm ± 0.0025 mm) in the range of 0.5 to 3 Hz was applied, as described previously [21,22]. The corresponding force ranged between 0 and 0.5 N. Displacement was controlled by a transducer with 15 nm resolution. The cyclic force and displacement were measured to obtain dynamic elastic (storage) (K′) and viscous (loss) (K′′) stiffness. Viscoelastic tangent delta (tan δ), which represents loading energy dissipation, was computed by K′′/K′. As such, the tan δ accounts for the relative capacity of dynamic energy dissipation [23,24].

### 2.4. Optical Image Scanning

Twenty-four hours after the last ML sessions, each mouse was injected retro-orbitally with 80 μL of solution containing a 1:1 mixture of MMPSense-750 (MMP750, PerkinElmer, Waltham, MA, USA) and monoclonal anti-type II collagen antibody (MabCII) that had been coupled with 680 dye (MabCII680). At 24 h after injection with the probe cocktail, mice were imaged under anesthesia after using an IVIS Imaging System (IVIS^®^ Lumina XR System, PerkinElmer, Waltham, MA, USA). Fluorescence was quantified using Living-Image 4.0 software (PerkinElmer, Waltham, MA, USA) to calculate the flux radiating omni-directionally from the region of interest (ROI) and graphed as radiant efficiency (photons/s/cm^2^/str)/(μW/cm^2^). The standardized ROI for knee fluorescence was obtained by capturing the same area for each mouse.

MabCII was provided by the laboratory of Karen Hasty at the VAMC (Memphis, TN, USA) [18,25,26]. The generation and characterization of MabCII have been previously described [26]. An immunoassay was performed to measure the binding and specificity of these antibodies for CII purified from mouse and other species. The results showed that MabCII (E4), the antibody chosen for this study, had the strongest immune reactivity.

The level of reactive oxygen species (ROS) and reactive nitrogen species (RNS) in the knee was measured by chemiluminescence after the first ML session, a useful time point because ROS are measurable immediately after loading, and after multiple sessions, the ROS/RNS signal can be detected systemically (Appendix A). Both knees were injected intra-articularly with 25 mg/Kg body weight L-012 sodium salt (TOCRIS, Bristol, UK) and immediately imaged by IVIS to determine the chemiluminescent signal [27].

### 2.5. Collection of Blood and Isolation of Cartilage Tissue

Blood (100 µL) was collected through the retro-orbital route using a sterilized heparinized capillary tube (Fisher Scientific, Hampton, NH, USA), and serum was isolated by centrifugation of blood (3000 rpm, 5 min). The animal was euthanized, and the entire knee joint was dissected to obtain cartilage tissues. The minced cartilage tissues were immediately transferred to RNAlater™ Solution (Thermo-Fischer, Waltham, MA, USA). The soft tissue was removed from the bones and the cartilage tissue was isolated from the femoral condyle and tibial plateau under a dissecting microscope.

### 2.6. Total Nitrate/Nitrite (NO) Assay

Serum was analyzed using the Nitrate/Nitrite Fluorometric Assay Kit, according to the manufacturer’s instructions (Cayman Chemical, Ann Arbor, MI, USA). The fluorescence signal was read at 375/417 nm (excitation/emission) using SpectraMax M5 microplate reader (Molecular Devices, Sunnyvale, CA, USA). The total NO was interpolated from the nitrate standard curve.

### 2.7. Caspase3 Assay

Caspase3 activity in serum was analyzed using the EnzChek^®^ Caspase-3 Assay Kit (Molecular Probes™, Eugene, OR, USA), according to the manufacturer’s instructions. The fluorescence signal was read at 496/520 nm (excitation/emission) using the microplate reader. The amount of fluorescence was used to determine the level of Caspase3 activity in serum.

### 2.8. Amplex Red Assay for H_2_O_2_

The level of H_2_O_2_ was measured in serum using the Amplex^®^ Red Hydrogen Peroxide/Peroxidase Assay Kit (Thermo-Fisher, Waltham, MA, USA). Fluorescence was read at excitation and emission wavelengths of 571 and 585 nm, respectively, using the microplate reader.

### 2.9. Prostaglandin E2 (PGE_2_) Assay

Serum was analyzed for PGE_2_ concentration with a Prostaglandin E2 Parameter Assay kit (Cayman Inc., Ann Arbor, MI, USA). The microtiter plate was read at a wavelength of 405 nm using a microplate reader.

### 2.10. Gene Expression Analysis

RNA was extracted from the cartilage tissue using the GeneJET RNA Purification Kit (Thermo-Fisher, Waltham, MA, USA). cDNA was prepared using 0.5 µg RNA by TaqMan^®^ Reverse Transcription Reagents (Thermo-Fisher, Waltham, MA, USA). To measure gene expression, reverse transcription-quantitative polymerase chain reaction (rt-qPCR) was performed using a TaqMan™ Assay (Thermo Fisher Scientific, Waltham, MA, USA) for the following genes: Aggrecan (ACAN), Collagen type II alpha (COL2A1), Endothelial PAS domain-containing protein 1 (EPAS1), Matrix metallopeptidase 13 (MMP13), Smad Nuclear Interacting Protein 1 (SNIP1), Interleukin 1 Beta (IL1β), and Tumor necrosis factor alpha (TNFα). Beta-Actin (β-Actin) served as an internal control. qPCR was performed on a LightCycler^®^ 480 (Roche, Basel, Switzerland). Data were analyzed using LightCycler^®^ 480 software (Roche, Basel, Switzerland) [20].

### 2.11. Histopathology (Modified OA and Synovitis Scoring System)

The knee joint was fixed in 10% formalin, decalcified with Decalcifying Solution (Thermo-Fischer, Waltham, MA, USA) for 7–10 days, and embedded in paraffin. A total of twenty-five histological sections per knee were analyzed to evaluate cartilage damage and synovitis. The sections were stained with hematoxylin and eosin stain (H&E) or Safranin-O/fast green stain.

Tissue sections were graded using the Osteoarthritis Research Society International (OARSI) and the Mankin scale with slight modifications [28]. A higher score represents a higher level of biological deterioration and disease extent, allowing for reliable differentiation between early- to late-stage OA. We further modified the procedure to assess the whole knee joint from an individual animal and to sum up the average histological score from each representative group. Briefly, the whole knee joint was divided into four areas: medial femoral condyle, medial tibial plateau, lateral femoral condyle, and lateral tibial plateau. Each area was graded based on hypocellularity, Safranin-O staining, surface regularity, and structure. An average histological score for each group of animals was obtained (Appendix A). Synovitis was graded as described previously with slight modifications [29]. The synovitis score was based on the enlargement of the synovial lining, density of cells, and inflammation.

### 2.12. Gait Analysis

Gait analysis was performed as described previously with slight modifications [30]. Hind and fore paws were stained with black and red nontoxic ink, respectively. Each animal was allowed to walk along a 100 × 10 × 15 (L × W × H cm) runway lined with white paper ending in a dark enclosed escape box. Animals were trained and acclimatized prior to recording the gait pattern.

### 2.13. Open-Field Analysis

Open-field analysis was performed using an Open Field Maze instrument (30 × 20 × 20 cm^3^) [31]. The animals were acclimatized to the Open Field Maze before the actual readings were made. Briefly, each animal was placed in the center of the instrument and allowed to move freely. The movement was recorded for thirty minutes. Any disturbances were avoided during the experiment. A ten-minute segment was analyzed using CowLog 3.0.2 software (Mattei Pastell, Helsinki, Finland) to determine the total travel distance and rearing activity [32].

### 2.14. Statistical Analysis

The significance of difference among more than two groups was determined by ANOVA followed by Bonferroni’s post-hoc multiple-comparisons test. Student’s *t*-test was performed to evaluate the significance of the difference between two groups. A *p*-value ≤ 0.05 was considered statistically significant. Data are presented as the mean ± standard deviation (SD). GraphPad Prism v.5.00 (GraphPad, San Diego, CA, USA) was used to perform statistical analyses.

## 3. Results

### 3.1. Mechanical Loading Induced Oxidative Stress, Unmasks Cartilage Type II Collagen, and Increases MMP Activity in Knee Joints

Oxidative stress was induced following the first ML session (Figure 1). The chemiluminescence signal increased significantly in the ML left knee joint as compared to the unloaded control right knee joint (4.46 ± 0.68 × 10^3^ ROI of ROS in control knee vs. 338.25 ± 88.33 × 10^3^ ROI of ROS in ML knee, *p* < 0.001), indicating that mechanical load increases both ROS and RNS.

We also confirmed the binding of MabCII680 to the damaged cartilage tissue in explant pig cartilage tissue and mouse knee joints (Figure 2). MabCII680 specifically binds to the damaged cartilage owing to mechanical damage and exposed type II collagen. An examination of mouse knee joints demonstrated that the intensity of MabCII680 and MMP750 signal increased in the traumatic ML joints relative to the unloaded control knee (1.47 ± 0.43 × 10^7^ ROI of MabCII680 in Normal vs. 19.34 ± 2.49 in Trauma vs. 23.52 ± 4.85 in PTOA knee joint, *p* < 0.001). While the MMP750 signal was significantly increased in the traumatic joint relative to the unloaded control knee joints, the signal then significantly decreased over time in the post-traumatic period (6.38 ± 1.19 × 10^5^ ROI of MMP750 in Normal knees vs. 37.94 ± 3.42 from the Trauma mice vs. 13.78 ± 5.51 in PTOA knee joints, *p* < 0.001).

### 3.2. Effect of Mechanical Loading on Biomechanical Properties

Biomechanical property alterations are one of the early signs of PTOA [29]. Therefore, the biomechanical features of ML knee joints were determined after conditions of acute trauma (Trauma group) or at 3 weeks after mechanical loading during the development of PTOA (PTOA group). Increases in the difference of viscoelastic tangent delta δ (tan δ) (1.0 ± 0.128 in Normal vs. 3.326 ± 0.331 in Trauma vs. 2.817 ± 0.311 in PTOA, *p* < 0.001) were observed relative to the control unloaded knee joints (Figure 3). A change in the mechanical properties of the cartilage/subchondral bone complex was observed based on the increased value of the tan δ.

### 3.3. Biochemical Changes Accompanied by Mechanical Loading

All biochemical analyses were performed using mouse serum from each specified experimental group. We measured total nitric oxide (NO) and found a significant increase in total NO in mice subjected to ML as compared to the untreated group (2790.77 ± 273.12 vs. 845.67 ± 110.94 pmoles, *p* = 0.01) (Figure 4A). We also measured caspase-3 levels in serum to determine if apoptosis might be driving pathology. We observed that ML caused a significant increase in caspase-3 activity as compared to the untreated control (370.65 ± 11.09 vs. 230.04 ± 6.37 fluorescence unit, *p* < 0.01) (Figure 4B). Hydrogen peroxide (H_2_O_2_) produced from oxidative stress can result in the destruction of tissues. We observed that H_2_O_2_ levels were significantly higher in the ML group relative to the untreated control group (8884.88 ± 92.06 vs. 8468.24 ± 118.59 fluorescence unit, *p* < 0.05) (Figure 4C). We also measured PGE2 levels that reportedly exert catabolic effects under mechanical stress. We observed that ML caused significantly high PGE2 levels in ML mice as compared to the untreated control (8015.62 ± 36.72 vs. 4627.50 ± 86.66 pg/mL, *p* < 0.01) (Figure 4D).

### 3.4. Mechanically Induced Gene Expression Patterns

Imbalances between anabolism and catabolism lead to pathogenesis of OA [33]. We studied the expression of both anabolic and catabolic genes. Expressions of two anabolism markers, major proteoglycan (ACAN) and collagen (COL2A1) were both reduced after ML relative to the untreated control (noML); ACAN (0.58 ± 0.03 vs. 1.00 ± 0.004-fold, *p* < 0.001) and COL2A1 (0.40 ± 0.10 vs. 1.01 ± 0.28-fold, *p* = 0.0245) (Figure 5A,B). In addition, the expression level of SNIP1, an inhibitor of NFκB signaling, decreased in ML mice relative to the control (0.24 ± 0.10 vs. 1.01 ± 0.27-fold, *p* = 0.0108) (Figure 5E). Genes responsible for cartilage catabolism increased after ML of the knee joint. EPAS1, a gene that encodes hypoxia-inducible factor (HIF-2α), increased significantly in the ML group relative to the control (6.17 ± 1.52-fold vs. 1.00 ± 0.14-fold, *p* = 0.0043) (Figure 5C). In addition, the expression of MMP13, a gene encoding a cartilage matrix degrading enzyme, also increased significantly in the ML group relative to the control (45.57 ± 1.11-fold vs. 1.04 ± 0.44-fold, *p* < 0.001) (Figure 5D). The inflammatory cytokines IL1β and TNFα were both enhanced significantly after ML as expected relative to control: IL1β (2.41 ± 0.19-fold vs. 1.00 ± 0.05-fold, *p* < 0.001) and TNFα (62.04 ± 1.21-fold vs. 1.00 ± 0.11-fold, *p* < 0.001) (Figure 5F,G). These inflammatory cytokines not only result in knee joint inflammation but also drive MMP13-mediated degradation of cartilage tissue.

### 3.5. Mechanical Loading of the Knee Joints Increased OA and Synovitis Score

Mechanical knee joint loading significantly impacted cartilage integrity (Figure 6A,B). The medial compartment was especially affected, possibly because the medial side bears more weight. Most prominently, a pronounced reduction in Safranin-O staining and a slightly irregular joint surface (Figure 6C,D) appeared in osteoarthritic joints. We also observed synovitis in knee joints subjected to ML (Figure 6E,F). The synovial membrane of osteoarthritic knees was highly inflamed relative to untreated controls. OA scores increased significantly in ML knee joints. Taken together, the histopathological observations clearly demonstrated signs of PTOA.

### 3.6. Altered Gait and Locomotory Behavior in Mechanically Loaded Mice

For a perspective on the functional consequences of the ML joint, we subjected mice to gait analysis. We observed that ML animals displayed a highly abnormal synchronous gait pattern relative to control mice (Figure 7A). Spatial variables including stride length, sway length (step width), and stance length describe the geometric position of the paw prints during locomotion. Rodents normally have symmetric gaits, where left and right limb foot-strikes (for either the fore or hind limbs) are spaced at approximately 50% of the gait cycle in time and equidistant in space (step length is about 50% of stride length). ML (both Trauma and PTOA group) reduced the stride length (−9.5% in Trauma and −4.9% in PTOA group), sway length (−6.1% in Trauma and −1.4% in PTOA group), and stance length (−12.3% in Trauma and −1.6% in PTOA group). Missing step and asymmetric gait patterns occurred with the unilateral injury. The missing step rates were increased in both Trauma (9.36%) and PTOA (4.64%) groups when compared to Normal control (Figure 7A and Table 1). Open-field analysis revealed a significant reduction in the total distance traveled (reduced average 26.69% in Trauma group and 30.68% in PTOA group when compared to Normal control) and rearing (reduced average 61.23% in Trauma group and 67.44% in PTOA group when compared to Normal control) following ML (Figure 7B,C). An evaluation of locomotor activity in the PTOA mice showed that the mean moving distance and rearing counts were significantly different from normal mice (Figure 7B,C).

## 4. Discussion

A murine model of PTOA helps investigators study the etiology of the early stages of OA [8,9]. We would propose that there are several distinct advantages of using the PTOA mouse model over the conventional model in which the medial meniscus is destabilized. The first is that the mechanical loading model is non-invasive and therefore avoids additional variables generated through destabilization surgery. A second advantage, also as a result of its non-invasive nature, is that it allows investigators to visualize the osteoarthritic changes occurring in the normal course of aging. Thirdly, the PTOA model mimics chronic OA and is helpful in predicting both early and late changes in the progression of the disease. As the changes in PTOA mouse model are progressive, it is obvious that the damage proceeds from the cartilage towards the subchondral bone. Our PTOA model employs a comprehensive approach including mechanical, biochemical, molecular, and functional changes. In this study, we used a specific mouse strain (CBA1) that we have successfully used in previous studies. While there is no reason to anticipate that other strains will behave differently, there are known differences in bone density that affect response to loading intensity and frequency. We are assuming that because they have different genetic backgrounds, we will preliminarily study how mechanical stress will affect the development of OA according to mouse strain. These effects are also influenced by age and sex. Pain, in particular, is one outcome that exhibits a differential response between males and females. By confining ourselves to male mice of a particular size and age, we reduced experimental variability and eliminated the need for extensive testing of the loading conditions necessary to produce PTOA of manageable severity. We utilized CBA-1 mice that have not shown any evidence of spontaneous OA [17,20,34], and therefore that would serve as an effective internal control. We found that the production of ROS/RNS was evident as early as the first ML session. The bioluminescence signal intensity was higher after the first ML session and declined over time (Appendix A). We observed increases in the levels of total NO and H_2_O_2_ in mechanically loaded mice, suggesting that ROS/RNS production plays a significant role in early PTOA. This finding suggests that studies focused on the role of oxidative stress in the development of PTOA in murine models should rely on early time points for ROS/RNS measurements [35,36].

We previously developed a MabCII680 that binds only to exposed CII in damaged cartilage [17]. We established that inflammation can be quantified through the measurement of MMP activity using the MMP750 probe [37]. In this study, we found increases in the signal intensity of both MabCII680 and MMP750 following ML. This suggests that cartilage damage and inflammation may be a consequence of mechanical load. Additionally, we found that mechanical stress compromises the biomechanical properties of articular cartilage. Structural and biochemical changes are associated with the pathogenesis of OA [38,39]. Changes observed in the viscoelastic properties can be explained due to an increase in the water content of the articular cartilage matrix with a concomitant decrease in proteoglycan levels and in CII content [40,41]. A previous study demonstrated changes in viscoelastic properties prior to any histologic signs of OA [42]. In contrast, we observed biomechanical property and histological changes.

Inflammation and apoptosis are well-established key players in the generation of OA [43,44]. We found that ML significantly enhanced joint inflammation. The PTOA mouse model demonstrated an increased serum production of PGE2 and the gene expression levels of IL1β and TNFα. Caspase-3 activity levels in serum also increased following ML. These data indicate that serum markers and articular cartilage gene expression analyses can both participate in the development of OA [33,45,46,47,48,49,50,51].

Histopathological analysis remains the gold standard in establishing the presence of OA in animal models [52]. In our model, we found that both the medial and lateral compartments were primarily affected. These compartments featured a significant loss of proteoglycans and an increased synovitis score. Meniscal damage was also prominent in the arthritic knee. We found multiple calcified regions with microtears in PTOA mouse knee joint. Thus, structural changes were vividly established in our PTOA model, consistent with the biomechanical, biochemical, and molecular alterations observed earlier.

We evaluated structural and biochemical changes in terms of the functional ability of mice affected by PTOA [53]. In our study, gait pattern and locomotor activity were severely affected in PTOA mice, highlighting the utility of functional outcome measurements in the PTOA model. Our observations correlated with a previous study showing that the ML of knee joints with 9N force produces lesions in ipsilateral and contralateral joints [54]. The authors indicated that the progress and worsening of lesions corresponded to the nociceptive behavior in loaded animals. It was emphasized that 9N force was more appropriate for longitudinal pharmacological studies due to mild joint damage. Our study agrees with these previous findings and provides insight into the behavioral changes due to the application of the 9N force for ML. Applying an optimal force is important in order to study the specific desired stage of disease. Our model showed that the application of 9N will allow us to measure functional changes associated with the application of a mechanical load. Post-loaded mice show lesions in the articular cartilage of the left knee, confirming that our ML protocol induces an OA-like histopathological phenotype. These results confirmed the spontaneous exacerbation of lesions at 3 weeks post-loading compared to directly after loading. PTOA mice appear to exhibit pain and attempt to walk with pain-related behavior. The timing of these lesions’ progression corresponds to the development of nociceptive behavior in this model, suggesting that progressive degradation of the knee induces these behavioral changes. Furthermore, the mild contralateral damage we found could explain the development of the contralateral mechanical hypersensitivity seen in these animals.

The results of CII targeted fluorescence and changes in the mechanical properties of the cartilage–bone complex for mechanically loaded mouse knees by DMA revealed that cartilage damage comes with changes in dynamic stiffness and loading energy dissipation of the complex. The results also suggest that these changes may affect the risk of damaging the subchondral bone from traumatic injury leading to OA later in life. In support of these early changes, Poulet et al. found that subchondral bone thickness increased in the mechanically loaded lateral compartment after five weeks of loading, predominantly in the femur over that of the tibia, and that the thickening was greater where load-induced articular cartilage lesions were present [55,56]. By contrast, epiphyseal trabecular thickness and mass increased in almost all knee joint compartments in the loaded mouse compared to joints from age-matched controls.

Interestingly, in previous studies, we found that different strains responded differently to the process of OA. For example, the DBA mouse legs are more easily fractured than other strains against the same mechanical loading condition, and the CBA mouse strain developed a more severe condition of OA than C57BL6 after mechanical loading.

## 5. Conclusions

Thus, ML increased oxidative stress, cartilage damage, subchondral bone change, and MMP levels in the affected knee joints [17,25,57]. The current findings will be useful as a monitoring tool for studying the progression of OA. This study helps correlate the severity of cartilage damage as shown by the OA score to the binding of MabCII to the cartilage and the change in viscoelastic tangent delta, representative of subchondral bone changes. These features can improve the objective assessment of various treatment and diagnostic OA studies. Furthermore, it strengthens the notion that studies focused on PTOA should be comprehensive in nature. Finally, the tools provided in this study can be utilized to better characterize murine PTOA based on biomechanical, biochemical, and behavioral parameters.

## Data Availability

Data will be shared with any biomedical research scientist who wishes to have it after the appropriate justification is supplied. The request must contain specific information about the nature of the data of interest, what the reason for the requester’s interest might be, and how the information will be used. The data to be released must all have been previously published more than one calendar year before the request.

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
