# Peer review of "Characterization of Non-Invasively Induced Post-Traumatic Osteoarthritis in Mice"

_antioxidants, 2022, doi:10.3390/antiox11091783_

Round 1
Reviewer 1 Report
The present study further characterized a murine model of non-invasive PTOA induction. The study is well written and interesting. To my mind, there are some minor aspects that would benefit from clarification or rephrasing, though.
Abstract: L26 “indicated increases in OA”, consider adding progression. L29, I am not sure if this last sentence is valid as such. Mouse models differ in several ways from human OA. Please consider rephrasing.
Intro: L66/67 although necrosis precedes apoptosis, this statement should be referenced; like Chen CT et al. JOR 2001.
Methods: is it correct that male 12-week-old mice were used in all experiments? L87, why did you use mice of this “particular age”? In line with this, Table 1 would benefit from explaining the differences between group 1 and 1’. Please mention the aspirated blood volumes (here or in Table 1). Please briefly explain how the cartilage was removed after samples were soaked in RNAlater. To my mind, #2.10. lacks important information: how did you extract 0.5 mg from mouse cartilage? You did not use qPCR, but RT-qPCR. Please correct. Please provide minimal essential information (primer assay info, RNA QC, amplification efficiency, normalization procedure and reference gene establishment) according to MIQE guidelines to ensure reproducibility. What was modified in the synovitis scoring with respect to Lewis et. al. (#27)? L199 please check unit, sounds odd.
Results: 3.1. Is there evidence that loading and/or oxidative stress “unmasks” type II collagen? Consider rephrasing, please. 3.3. Why are two Refs provided in L259? L262/3 moving all Refs to Discussion? Fig. 5; Y-axes largely differ, raising questions about the normalization strategy. See also my point to #2.10.
Discussion: I suggest moving L433 following up to 355 f., where strain- and age-specific limitations are discussed. As another limitations, a sentence towards differences between OA in mice and men may be added.
You used a non-invasive PTOA model (#2.2.), referring to refs 7-19. Which aspects were slightly modified (L98)? In Fig. 6, Ac shows what looks like a posterior crucial ligament rupture. Is that representative or a preparation artifact? Are all top images in anterior-posterior view?
Author Response
Reviewer 1]
The present study further characterized a murine model of non-invasive PTOA induction. The study is well written and interesting. To my mind, there are some minor aspects that would benefit from clarification or rephrasing, though.
Abstract: L26 “indicated increases in OA”, consider adding progression. L29, I am not sure if this last sentence is valid as such. Mouse models differ in several ways from human OA. Please consider rephrasing.
Response:
We rewrote the L26 as:
Histopathology scores clearly indicated increases in OA progression and synovitis.
We rewrote the L29 as:
These results validate that the current mouse model can be considered for translational PTOA studies.
Intro: L66/67 although necrosis precedes apoptosis, this statement should be referenced; like Chen CT et al. JOR 2001.
Response:
We added the following reference according to reviewer’s comment:
Chen CT, Bhargava M, Lin PM, et al. 2003. Time, stress, and location dependent chondrocyte death and collagen damage in cyclically loaded articular cartilage. J Orthop Res 21:888–898.
Methods: is it correct that male 12-week-old mice were used in all experiments? L87, why did you use mice of this “particular age”? In line with this, Table 1 would benefit from explaining the differences between group 1 and 1’. Please mention the aspirated blood volumes (here or in Table 1). Please briefly explain how the cartilage was removed after samples were soaked in RNAlater. To my mind, #2.10. lacks important information: how did you extract 0.5 mg from mouse cartilage? You did not use qPCR, but RT-qPCR. Please correct. Please provide minimal essential information (primer assay info, RNA QC, amplification efficiency, normalization procedure and reference gene establishment) according to MIQE guidelines to ensure reproducibility. What was modified in the synovitis scoring with respect to Lewis et. al. (#27)? L199 please check unit, sounds odd.
Response:
Mice at twelve weeks of age are classified as mature from a musculoskeletal perspective and should be considered as young adults in comparison to humans.
We add this description in Table 1: Group 1 and 1’ is a normal control but at a different age (i.e.: Group 1 is normal at 12 weeks of age and group 1’ is normal at 19 weeks of age)
100ul Retroorbital blood collection using a capillary glass tube: added in the table 1 and in section #2.5.
L136: We rewrote as a “Blood (100ul) was collected through retro-orbital route using a sterilized hepa-rinized capillary tube (Fisher Scientific, Hampton, NH), and serum was isolated by centrifugation of blood (3,000 rpm, 5 min)”.
L138: We agree to reviewer’s comment. We rewrote more carefully in #2.5 section (Collection of blood and isolation of cartilage tissue): as “The animal was euthanized and the entire knee joint was dissected to obtain cartilage tissues. The minced cartilage tissues were immediately transferred to RNAlater™ Solution (Thermo-Fischer, MA).”
This method usually yields sufficient RNA from the knee joint to provide the 0.5 µg of RNA for rt-PCR.
We appreciate the correction. We used rt-qPCR and have documented this as:
- “To measure gene expression, reverse transcription-quantitative polymerase chain reaction (rt-qPCR) was performed using a TaqMan™ Assay (Thermo Fisher Scientific, Waltham, MA) for the following genes:”
For more information, please see our previous publication: Cho H., Walker A., Williams J., Hasty K.A. “Study of Osteoarthritis Treatment with Anti-inflammatory Drugs: Cyclooxyganase-2 (COX2) Inhibitor and Steroids” BioMed Research International 2015 Article ID 595273, 10 pages (PMID: 26000299)
Results: 3.1. Is there evidence that loading and/or oxidative stress “unmasks” type II collagen? Consider rephrasing, please. 3.3. Why are two Refs provided in L259? L262/3 moving all Refs to Discussion? Fig. 5; Y-axes largely differ, raising questions about the normalization strategy. See also my point to #2.10.
Response:
In previous studies, we have shown that the MabCII680 only binds to damaged cartilage surfaces.
Ref: Cho H., Pinkhassik E., David V., Stuart J.M., Hasty K.A. “Detection of Early Cartilage Damage using Targeted Nanosomes in a Post-Traumatic Osteoarthritis Mouse Model” Nanomedicine: NBM 2015; 11: 939-946 (PMID: 25680539)
This change is reflected as: "MabCII680 specifically binds to the damaged cartilage owing to mechanical damage and exposed type II collagen".
We removed the citations from the Results section and added them to the Discussion section as per the reviewer’s suggestion.
Discussion: I suggest moving L433 following up to 355 f., where strain- and age-specific limitations are discussed. As another limitations, a sentence towards differences between OA in mice and men may be added.
Response:
The paragraph has been moved to after L355. We appreciate the reviewer’s suggestion.
You used a non-invasive PTOA model (#2.2.), referring to refs 7-19. Which aspects were slightly modified (L98)? In Fig. 6, Ac shows what looks like a posterior crucial ligament rupture. Is that representative or a preparation artifact? Are all top images in anterior-posterior view?
Response:
We modified the duration and frequency of mechanical loading to induce mild damage rather than ACL rupture. These features also distinguished it from the DMM model. The 9N compressive force is the same as in the original model by Dr. Poulet's group. Another difference is that the inside of the jig engineered to hold the mouse leg for mechanical loading has a more rounded shape than the original jig. Our peak loads of 9N were applied for 1 seconds, with a rise and fall time each of 1 seconds and a baseline hold time of 1 seconds to mimic a fast walking or jogging condition.
Our loading protocol does not induce a severe condition such as a posterior crucial ligament rupture. This reflects an artifact of damage due to sample preparation, and we did not use that area for scoring.
Reviewer 2 Report
This is an interesting study investigating the use of ML-induced OA in a mouse model. The authors used different experimental approaches to characterise the biochemical and functional changes after ML and concluded that this PTOA mouse model could be useful for studying the progression and different pathological features of OA.
The authors need to address the following points:
1. Abstract, lines 25-26. The authors should state the specific genes analysed.
2. Line 43, replace “tissues” with components.
3. Lines 78-79, the behavioral data were indeed analysed in this study (gait etc) but how pain was assessed? This has to be clearly linked and supported by the literature.
4. Line 82, n=16/group. Some joints from each group (n=6 according to legends) were fixed for histology, other samples were used for RNA extraction, other for DMA etc. This is confusing. Please clarify and clearly state when a knee joint was used for different measurements. In addition, all graphs should be presented as scatter plots with bars.
5. Line 102, what does every other day mean? How many times per week? This needs clarification and the loading regime has to be clear.
6. Section 2.5. How was harvest of the cartilage achieved without a microscope?
7. Section 2.11. Since the authors used a modified scoring system incorporating different hallmarks of OA, representative histological pictures must be provided in the supplemental material for the different scores in the different categories (FigS4). For example, it would be useful to show three sections, one from each group, with the detailed scores. Also, a summative table for all scores must be included in the supplemental material.
8. Line 216-17. Is this 338.25 ± 88.33 x 103? It needs to be corrected here and in the next paragraph, lines 230 and 234.
9. Sections 3.1. and 3.2. and Figs 2 and 3 can be merged.
10. Overall, in the Results part, the authors should clearly state the time point and the tissue (cartilage, serum etc) for each measurement as well as which experiments took place in vivo and which ex vivo.
11. Lines 259-260. “We also measured Caspase-3 levels in the joint to determine if apoptosis might be driving pathology”. How was Caspase-3 measured in the joint since serum samples were utilised?
12. Fig6. It does not seem that the magnified H&E images are from the same sections as stated in the legend. Are these rom other sections? Please comment. The authors state that “Cell distribution is affected by cartilage damage”. This should be indicated with arrows. Have the authors observed other OA hallmarks like PG deposition in the collateral ligament? Please comment. Why was synovial thickening not incorporated in the scoring system used in this study? Finally, IHC for MMP13 is essential to show early Col2 degradation and ADAMTS/MMP-derived neoepitope detection would be useful as well to show aggrecan cleavage.
13. Line 391. The authors state that changes were observed in the medial compartment. However, in Fig6A it is clearly shown that PG loss can be observed in all four compartments of the joint in the PTOA group stained with SafO and also cartilage lesions are obvious in the LF of the same group when stained with H&E. In contrast, Poulet et al. [18] has reported that using the same ML model in the same mouse strain this is observed only in LF. Please comment.
14. Line 393. The authors wrote that “Meniscal damage was also prominent in the arthritic knee”. Where is the evidence for this?
15. Lines 410-12. The time points between this study and [18] are not the same. The authors need to amend or delete this statement.
Author Response
Reviewer 2]
This is an interesting study investigating the use of ML-induced OA in a mouse model. The authors used different experimental approaches to characterise the biochemical and functional changes after ML and concluded that this PTOA mouse model could be useful for studying the progression and different pathological features of OA.
The authors need to address the following points:
- Abstract, lines 25-26. The authors should state the specific genes analysed.
Response:
We added the specific gene name and rewrote as a “Gene expression analysis highlighted increased catabolism (MMP-13, IL-1β, TNF-α) with concomitant decrease in anabolism (ACAN, COL2A1).”
- Line 43, replace “tissues” with components.
Response:
We have rewritten as “This PTOA is characterized by damage to multiple components such as articular and meniscal cartilages, synovium and subchondral bone.”
- Lines 78-79, the behavioral data were indeed analysed in this study (gait etc) but how pain was assessed? This has to be clearly linked and supported by the literature.
Response:
The progression of OA is accompanied by secondary clinical symptoms, loss of mobility and altered gait caused by pain. To confirm this condition, we performed open field analysis for indirect measurement of the pain related behavior.
At 3 weeks following non-invasive mechanical loading (ML: 9N), PTOA mice exhibited a reduction in the distance traveled and rearing while moving relative to controls without ML. These results showed that mechanical stress reduced the locomotor behavior (i.e., distance traveled and rearing) compared to the normal mice, an indirect indicator of OA-related pain in PTOA mice in Fig 7. This result will be compared to more direct measurement like a von Frey testing in a future study.
- Line 82, n=16/group. Some joints from each group (n=6 according to legends) were fixed for histology, other samples were used for RNA extraction, other for DMA etc. This is confusing. Please clarify and clearly state when a knee joint was used for different measurements. In addition, all graphs should be presented as scatter plots with bars.
Response:
The samples were divided into three different preparations according to analysis:
For histopathology, the specimens (6) were placed in 4% para-formaldehyde
For RNA extraction, the dissected knee cartilage (5) were placed in RNAlator
For DMA, whole knee joints (5) were frozen until needed.
All experiments were performed in triplicate and the specimens in each group were divided into five different analyses for histopathology, biochemistry, RNA extraction, bone morphology (μCT) and pain related behavior.
- Line 102, what does every other day mean? How many times per week? This needs clarification and the loading regime has to be clear.
Response:
We agree the description was not clear. We have rewritten it as: “Each loaded knee joint received 40 cycles of compressive loading at 9 N, totaling six successive sessions (3 times per week over 2 weeks).
- Section 2.5. How was harvest of the cartilage achieved without a microscope?
Response:
We dissected the knee joint under a dissecting microscope. Rewritten as: The soft tissue was removed from the bones and the cartilage tissue was isolated from the femoral condyle and tibial plateau under a dissecting microscope.
- Section 2.11. Since the authors used a modified scoring system incorporating different hallmarks of OA, representative histological pictures must be provided in the supplemental material for the different scores in the different categories (FigS4). For example, it would be useful to show three sections, one from each group, with the detailed scores. Also, a summative table for all scores must be included in the supplemental material.
Response:
We provide additional slides and score tables in the Supplement section (Supplement Figure 4 D).
- Line 216-17. Is this 338.25 ± 88.33 x 103? It needs to be corrected here and in the next paragraph, lines 230 and 234.
Response:
We all appreciate that the reviewer found this typo The chemiluminescence signal increased significantly in the ML left knee joint as compared to unloaded control right knee joint (4.46 ± 0.68 ×103 ROI of ROS in control knee vs 338.25 ± 88.33 ×103 ROI of ROS in ML knee, p < 0.001), indicating mechanical load increases both ROS and RNS.
- Sections 3.1. and 3.2. and Figs 2 and 3 can be merged.
Response:
We are understanding the reviewer’s concern. However, there were totally two different assays performed in 3.1 and 3.2.
Figure 2 is biological measurement whereas Figure 3 is for biomechanical measurement. We therefore believe separate figures more accurately reflect what was measured.
- Overall, in the Results part, the authors should clearly state the time point and the tissue (cartilage, serum etc) for each measurement as well as which experiments took place in vivo and which ex vivo.
Response:
We have indicated all of the serum-based biochemical analyses in the Methods section. To make this more clear, we have added the following sentence in #3.4:
“All biochemical analysis were performed using mouse serum from each specified experimental group”.
- Lines 259-260. “We also measured Caspase-3 levels in the joint to determine if apoptosis might be driving pathology”. How was Caspase-3 measured in the joint since serum samples were utilised?
Response:
We now recognize that the description in the manuscript is not correct.
We measured Caspase-3 levels in serum, and not in the specific knee joint tissue. We have previously tried to measure knee joint Caspase-3 using synovial lavage fluids but were unsuccessful.
We have revised the corresponding paragraph as follows:
We also measured Caspase-3 levels in serum to determine if apoptosis might be driving pathology. We observed that ML caused a significant increase in caspase-3 activity as compared to untreated control (370.65 ± 11.09 vs 230.04 ± 6.37 flu-orescence unit, p < 0.01) (Figure 4B).
- Fig6. It does not seem that the magnified H&E images are from the same sections as stated in the legend. Are these rom other sections? Please comment. The authors state that “Cell distribution is affected by cartilage damage”. This should be indicated with arrows. Have the authors observed other OA hallmarks like PG deposition in the collateral ligament? Please comment. Why was synovial thickening not incorporated in the scoring system used in this study? Finally, IHC for MMP13 is essential to show early Col2 degradation and ADAMTS/MMP-derived neoepitope detection would be useful as well to show aggrecan cleavage.
Response:
- For clarity purposes, we used different sections of the magnified images. We used at least 4 different sections for scoring and one of those images was represented in these magnified images.
- We indicated the cell distribution change with yellow arrow head in this figure 6 A c.
-We adopted the synovitis scoring system from the following reference:
Lewis JS, Hembree WC, Furman BD, Tippets L, Cattel D, Huebner JL, Little D, DeFrate LE, Kraus VB, Guilak F, Olson SA. Acute joint pathology and synovial inflammation is associated with increased intra-articular fracture severity in the mouse knee. Osteoarthritis Cartilage. 2011 Jul;19(7):864-73. doi: 10.1016/j.joca.2011.04.011. Epub 2011 May 12. PMID: 21619936; PMCID: PMC3312469.
-In our previous studies we have used IHC to demonstrate that MMP-13 is increased in a mechanically loaded knee joint. Please see the following references.
Refs]
Bedingfield S., Colazo JM, Yu F, Liu DD, Jackson MA, Himmel LE, Cho H., Crofford L, Hasty K., Duvall CL. “Amelioration of post-traumatic osteoarthritis via nanoparticle depots delivering small interfering RNA to damaged cartilage” Nature Biomedical Engineering 2021 Sep;5(9):1069-1083.. (DOI: https://doi.org/10.1038/s41551-021-00780-3)
O'Grady K, Kavanaugh T., Cho H., Ye H., Gupta M., Madonna M, Lee J., O'Brien C., Skala M., Hasty K., Duvall CL. "Drug Free ROS Sponge Polymeric Microspheres Reduce Tissue Damage from Ischemic and Mechanical Injury" ACS Biomaterials Science & Engineering 2018, 4(4): 1251–1264 (DOI: 10.1021/acsbiomaterials.6b00804) (PMID: 30349873, PMCID: PMC6195321)
We agree with reviewer's concern and we will prepare new manuscript about more specific histopathological characterization containing the IHC in the near future.
- Line 391. The authors state that changes were observed in the medial compartment. However, in Fig6A it is clearly shown that PG loss can be observed in all four compartments of the joint in the PTOA group stained with SafO and also cartilage lesions are obvious in the LF of the same group when stained with H&E. In contrast, Poulet et al. [18] has reported that using the same ML model in the same mouse strain this is observed only in LF. Please comment.
Response:
In our previous study, we found that both of the medial and lateral compartment was affected. We hypothesize that this difference may be due to differences in the shape of our custom leg jig. The inside of the jig designed to hold the mouse leg for mechanical loading has a round shape rather than the flat shape found in the original jig. Because of this shape-change, the contact points may be varied.
We appreciate that this was pointed out by Reviewer 2 and have rewritten the section thusly:
In our model, we found that both of the medial and lateral compartment were primarily affected. This compartment featured a significant loss of proteoglycans and an increased synovitis score. Meniscal damage was also prominent in the arthritic knee. Thus, structural changes were vividly established in our PTOA model, consistent with the biomechanical, biochemical, and molecular alterations observed earlier.
- Line 393. The authors wrote that “Meniscal damage was also prominent in the arthritic knee”. Where is the evidence for this?
Response:
We found multiple calcified regions with microtears in PTOA mouse knee joint histology slides.
We have added multiple H&E images in the Supplement section (please see the Supplement Figure 4D).
- Lines 410-12. The time points between this study and [18] are not the same. The authors need to amend or delete this statement.
Response:
We agree with reviewer's concern that the pattern of loading regimen is similar, but the applied time duration or time point are not same. We removed the statement according to reviewer’s comment.
Round 2
Reviewer 2 Report
The authors efficiently addressed my points. This is an interesting work and the current revised form of the manuscript is publishable.